# Steryl Glycosides in Fungal Pathogenesis: An Understudied Immunomodulatory Adjuvant

**DOI:** 10.3390/jof6010025

**Published:** 2020-02-24

**Authors:** Tyler G. Normile, Kyle McEvoy, Maurizio Del Poeta

**Affiliations:** 1Department of Microbiology and Immunology, Stony Brook University, Stony Brook, NY 11794, USA; tyler.normile@stonybrook.edu (T.G.N.); kyle.mcevoy@stonybrook.edu (K.M.); 2Division of Infectious Diseases, School of Medicine, Stony Brook University, Stony Brook, NY 11794, USA; 3Veterans Administration Medical Center, Northport, New York, NY 11768, USA

**Keywords:** sterols, steryl glycosides, lipid metabolism, fungal pathogenesis, host immunity, vaccine candidates, immunomodulatory, adjuvants

## Abstract

Invasive fungal infections pose an increasing threat to human hosts, especially in immunocompromised individuals. In response to the increasing morbidity and mortality of fungal infections, numerous groups have shown great strides in uncovering novel treatment options and potential efficacious vaccine candidates for this increasing threat due to the increase in current antifungal resistance. Steryl glycosides are lipid compounds produced by a wide range of organisms, and are largely understudied in the field of pathogenicity, especially to fungal infections. Published works over the years have shown these compounds positively modulating the host immune response. Recent advances, most notably from our lab, have strongly indicated that steryl glycosides have high efficacy in protecting the host against lethal Cryptococcal infection through acting as an immunoadjuvant. This review will summarize the keystone studies on the role of steryl glycosides in the host immune response, as well as elucidate the remaining unknown characteristics and future perspectives of these compounds for the host–fungal interactions.

## 1. Introduction

Common to all kingdoms of living organisms are the four classes of biological macromolecules: nucleic acids, proteins, carbohydrates, and lipids. The latter of these biomolecules has become a major topic of investigation in several fields of biology, including cell structural studies, metabolomic and signaling studies, industrial uses, and dictating virulence of pathogens towards both plants and animals [1,2,3,4,5,6,7,8,9]. Lipids are broadly defined as a diverse class of hydrophobic molecules that take part in a multitude of biological functions, most notably in eukaryotic cells. There are several classes of lipids, including triacyl glycerides, sterols, sphingolipids, phospholipids, glycerophospholipids, and glycolipids. Whether or not certain species of lipids are produced in a cell will vary by the organism, whereas certain lipid species will vary by the kingdom, the genus, or even the strain of an organism [10,11,12]. Indeed, the lipidome, or the totality of all the lipids of a cell, has been an evolving topic of interest for plants, fungi, yeasts, and mammals, including both rodents and humans [13]. To add to the complexity and diversity of the lipidome, certain lipids can be decorated with different numbers and types of carbohydrate moieties for varying physiological effects, such as the case with ceramides to glucosylceramides, as well as sterols to steryl glycosides (SGs) and acyl-SGs. The majority of the published work on SGs has been in relation to plants, both from the host and pathogen sides [11,14,15,16], since plant pathogens take a serious monetary toll on the agricultural industry and also can be used to induce more robust plant immune resistance to pathogens [8,17]. However, SG metabolism has been an understudied topic in fungal pathogenesis with regards to the mammalian host relative to other compounds and virulence-associated factors commonly found in the literature. This review will cover a broad introduction to SG metabolism and aims to primarily focus on recently published work concerning SGs and their immunological role in fungal pathogenesis in the mammalian host as well as other applications found in the literature. 

## 2. Lipids as An Emerging Topic of Study in Fungal Infections

The investigation into lipids, specifically in fungi, has gained interest due to the recent emergence of fungi as human pathogens over the past few decades as well as being problematic to the agricultural industry [7,18,19]. The number of cases of invasive fungal infections have skyrocketed with the growing population of immunocompromised individuals, most notably including AIDS patients and patients undergoing immunosuppressive medical interventions. Invasive fungal infections kill an estimated 1.5 million people worldwide each year with the species *Cryptococcus*, *Aspergillus*, *Candida*, and *Pneumocystis* resulting in the majority of deaths [20,21], but several other endemic fungal genera also contribute to the growing mortality rate, including *Histoplasma*, *Paracoccidioides*, *Blastomyces*, and *Coccidioides*. Currently, the antifungal treatment options are lackluster at best and, since fungi belong to the eukaryotic domain of life, the development of treatment options is more confined so as not to harm the host as well, unlike the availability of antibiotics to bacteria. Indeed, the most common antifungal treatment options currently available target the plasma membrane lipid components of these pathogens, such as the azole drugs that target ergosterol synthesis and the polyenes that bind to sterols [22,23]. However, acquired resistance, price and availability, narrow spectrum of activity, and host toxicity are major problems with these treatments and thus highly justify the search for novel anti-fungal treatments and targets, especially with the increased number of immunocompromised patients.

In addition to the known structural and regulatory roles, lipids have also been described to be virulence factors in pathogenic fungi. Due to the availability of more precise assays, unique structural differences in fungal lipid species compared to the host have been identified, which have opened the possibility of exploring these compounds and their associated enzymes as novel antifungal targets [10,18,24,25]. Specific lipid species that have been suggested to be potential novel drug targets include cell membrane-associated lipids such as phospholipids and glycosphingolipids. In eukaryotes, phosphatidylcholine (PC) and phosphatidylethanolamine (PE) represent the most abundant glycerophospholipids produced by the cytidine diphosphate diacylglycerol (CDP-DAG) pathways and/or the Kennedy pathway in mammalian cells, parasites including *Trypanosoma brucei*, and fungi including *Candida albicans,* among others [26,27,28]. Production of PE and phosphatidylserine (PS) via the CDP-DAG pathway are necessary for pathogenesis in vivo during a systemic *C. albicans* infection [27]. Moreover, disruption of the Kennedy pathway via genetic knockout mutant strains leads to attenuated virulence and, conversely, a mutant strain overexpressing EPT1 (ethanolamine/cholinephosphotransferase) was hypervirulent in a mouse model. These data shed light onto the regulation of virulence via phospholipids in fungi. For further information, we refer the readers to more recent comprehensive reviews on phospholipids [29,30] and the Kennedy pathway [31] as another source of lipid research for drug targets, since the remainder of this review will focus on glycosphingolipids, namely steryl glycosides.

Several reports have established certain enzymes, including inositol phosphosphingolipid-phospholipase C1 (Isc1), glucosylceramide synthase 1 (Gcs1), and sterylglucosidase 1 (Sgl1), as key virulence factors in *Cryptococcus neoformans*, signifying that such glycosphingolipids as inositol sphingolipids, glucosylceramide, and steryl glucosides can regulate fungal pathogenicity, respectively [32,33,34]. More specifically, *C. neoformans* Δ*isc1* mutants have strongly attenuated virulence in vivo via defective growth in the phagolysosomal compartments of host phagocytes upon infection [34]. *C. neoformans* Δ*gcs1* mutants cannot survive in the extracellular environment of the host, such as the alveolar space and bloodstream, rendering this mutant avirulent in a mouse model of infection and the first microbial lipid to be identified as a virulence factor [32]. More interestingly, however, *C. neoformans* Δ*sgl1* mutants accumulate steryl glucosides with no other obvious virulence factor defects (Figure 1), which results in a robust immune response by the host clearing the yeast from the lungs by day 14 [33]. Most intriguing is the complete protection conferred to the host upon subsequent challenge with the virulent H99 strain in both immunocompetent and immunocompromised hosts. These clinically relevant observations strongly advocate that future investigation into SGs and host protection may be advantageous for potential antifungal vaccine therapies. The remainder of this review will focus on relevant work regarding SGs and the sterol derivatives with respect to pathogenic microorganisms, with an emphasis on pathogenic fungi in mammals and plants as well as several other less studied applications of SGs.

## 3. Sterols and Steryl Glycosides

### 3.1. Sterols

Sterols are lipid compounds that are ubiquitously found in the plasma membranes of plants, fungi, mammals, among other eukaryotic organisms, and play a role in the regulation of membrane fluidity, permeability, and lipid raft formation due in part to the aliphatic side chains [12,17,35,36]. The type of major sterol species varies depending on the organism, whereas cholesterol is found in the plasma membranes of animal cells, and plants contain a combination of sitosterol, stigmasterol, and campesterol [15,17]. Finally, ergosterol is the major sterol species required for fungal growth, protein folding, and cell cycle regulation and is found in the fungal plasma membranes in both plant and animal pathogens [10,37,38]. For a full characterization, organization, function, and backbone, fatty acid, and headgroup composition, we refer the readers to the review of steryl glycosides by Grille et al. [15] and lipid compounds by Nimrichter et al. [7], who have fully described all of the relevant compounds, which will not be repeated here. Although ergosterol was found to be the most abundant sterol species in pathogenic fungi such as *C. neoformans* and *C. albicans,* as well as in non-pathogenic *Saccharomyces cerevisiae* [10,12], some fungi do show variations in sterol abundance, such as in the subphyla Pucciniomycotina [17]. Furthermore, the oomycete pathogens in the genera *Pythium* and *Phytophthora* require an external source of sterols and are believed to be from the host cell membrane. The characterization of these sterols led to the development of many antifungals that target the ergosterol pathway in fungi, including the allyamines and the azoles targeting key ergosterol pathway enzymes, and the polyenes that are pore-forming compounds that bind ergosterol [17,39,40]. From the plant side, plants have also adapted ways to sense and respond to fungal sterols including ergosterol [17]. Avenacin is secreted by plants such as oat roots to form membrane altering complexes in the fungal pathogens. With the importance of ergosterol in fungal pathogenesis, understanding ergosterol and other sterols as immunological targets has been of interest and will be briefly discussed.

Sterols have been implicated in both inflammatory and anti-inflammatory roles. Cholesterol was shown to be involved in the inflammatory and innate immune response [41,42]. Indeed, depletion of cholesterol from macrophage cell membranes affects the host’s phagocytic efficacy of *C. neoformans*. Additionally, free cholesterol functions in part as a ligand for toll-like receptors on macrophages, activation of inflammasome, and myelopoiesis inducing a pro-inflammatory response in the host [41]. Hence the regulation of cholesterol is important for the proper host response to a pathogen, where too little or much is detrimental to the outcome. However, pathogen-mediated glycosylation of host cholesterol, mainly by bacterial pathogens, including *Helicobacter pylori* and *Borrelia burgdorferi*, was shown to be harmful to the host by affecting the associated immune response, which will be discussed in more detail in the bacterial SG section below [43,44,45,46].

On the fungal side, ergosterol is a fungal pathogen-associated molecular pattern (PAMP) in both animal and plant pathogens, although chemically and structurally similar to both cholesterol in animals and sitosterol in plants [37,47]. Ergosterol was discovered about a century ago in the plant pathogen *Claviceps purpurea* but is not present in all fungi, as mentioned previously [12,17]. Ergosterol dominates the Ascomycota and Basidiomycota phyla, which encompass the more medically relevant fungi where ergosterol is the dominant sterol species. On the same note as cholesterol, cellular localization and concentration of ergosterol in *C. neoformans*, *C. albicans*, and *Saccharomyces cerevisiae* were shown to trigger pyroptosis in macrophages in a keystone study screening deletion mutants [9]. Additionally, an accumulation of sterols was found in the tips of budding and hyphal growing cells of *C. neoformans*, *Aspergillus fumigatus*, and *C. albicans* [48,49,50,51,52,53], indicating the role of sterol conglomeration at these division points. Ergosterol was also found to be necessary for biofilm formation in *C. albicans* [54,55]. Although ergosterol levels dropped to roughly half after 48 hours, the effects of simvastatin were still abrogated by addition of exogenous ergosterol, which may suggest a titrating effect of the exogenously added ergosterol and/or the necessity of ergosterol early on in biofilm formation but not necessary for retention.

There have also been several attempts at the production of antibodies against ergosterol. Just over two decades ago, Tejada-Simon and Pestka [56] attempted to produce a polyclonal antibody against ergosterol hemisuccinate. In the end, there was successful binding of the antibodies to the ergosterol–hemisuccinate–BSA conjugate, but none of the antibodies bound free ergosterol. This is most likely due to a binding epitope found on the conjugate but not ergosterol alone. In the end, an antibody binding specifically to ergosterol would highly benefit studies aimed at studying sterol metabolism, localization, and turnover rate in fungi. 

Since the sterol pathway is a common drug target in pathogenic fungi, changes in sterol composition have been explored in the past. Ghannoum and colleagues [57] investigated how the sterol composition in *C. neoformans* changed in clinical isolates of relapse patients compared to primary infection patients, unlike what was seen in *C. albicans*. Although this study was published over two decades ago, a major takeaway from this work stems from the idea that the sterol composition changes upon treatment with fluconazole (a drug targeting a specific enzyme in the ergosterol pathway), so treatment options for patients with relapse cryptococcosis may need to be altered. This also signifies a large need for altered drug treatment options that are not centered specifically around targeting ergosterol. The following section will cover important findings and data associated with the decorated alterations of these sterols including glycosylation and acylation. 

### 3.2. Steryl Glycosides and Other Conjugated Sterols

Sterols may be decorated with sugar moieties of differing number or species as well as other acyl modifications to alter their structure and functional roles in the plasma membrane and internalized compartments such as lysosomes and tonoplasts in plants [1,6,7,11,15,58,59,60,61]. SGs are structurally defined as the condensation reaction of donated sugar derivative to the 3-hydroxyl group of a free sterol. Additionally, most of the SGs have a β-linked sugar addition, while ⍺-linkages also exist in vivo but to a lesser extent. These modifications change the size, viscoelastic behavior, and permeability of the sterol as well as the biological membrane. It was shown by Chang and colleagues [62] that the glucosyl group addition was critical for uptake of SGs into *C. albicans* in a study investigating the minimum inhibitory concentration (MIC) of exogenously added SGs. They also found that there was ~10-fold increase in membrane glucosidase activity if glucose was the monosaccharide addition versus other sugar derivatives. Indeed, currently much more is known regarding the biophysical properties of these altered sterols, but research about how these alterations change the downstream biological functions within the cells is still lacking, although our lab has recently made great strides in uncovering some immunological roles of SGs in the human fungal pathogen *C. neoformans* [2,33]. While assessing the literature regarding SGs, knowledge of how they are synthesized via steryl glycosyltransferase enzymes, and how they are catabolized via sterylglucosidase enzymes (shown in Figure 1) is necessary since both making and breaking of SGs happens very rapidly in an organism and much of the literature focuses on one side or the other. 

SGs are known to be found in plants, algae, fungi, and yeasts, although varying concentrations, strain specificity, and stress-mediated responses play large roles in the intensity of effects reported in the literature [11,15,33,58,61]. It was found by Sakaki and colleagues [61] that not all fungi make SGs, but growth conditions and media choices highly influence SG production, since several fungal species that possess an active steryl glycosyltransferase enzyme do not have quantifiable SGs. In more detail, two different strains of *Pichia pastoris* and one strain of *Sordaria macrospora* were shown to all produce β-D-glucopyranoside, but the two different strains of *P. pastoris* were shown to produce differing levels of SGs and also be dependent on the type of stressor. Furthermore, little was known about the function of SGs in animals and bacteria as most of the research has been focused in plants and fungi, although knowledge regarding SG metabolism in mammals and bacteria is beginning to open up [44,45,59,63]. Overall, SGs are not one compound but an umbrella term used to describe the addition of a variable sugar moiety to a sterol species, so the diverse effects reported in the literature is justified. The major topics of study regarding SGs is depicted in Figure 2. The majority of the studies utilize plant-derived SGs for immunological studies, cancer studies, cloning, and biotechnological uses, however, fungal, mammalian, and bacterial SGs are also included in at least one or more studies each. While each of the functions will be discussed in more detail in the upcoming sections, major differences in experimental approaches are used throughout, which may be a vital reason for differences in efficiency of SGs among studies. 

Finally, much of this research came to fruition due to the first identification, description, and cloning of the steryl glycosyltransferase (SGT) and sterylglucosidase (Sgl) enzymes in fungi. Both SGT and Sgl enzymes were described in plants, fungi, yeast, and more recently bacteria, humans, and rodents [1,8,11,15,33,44,46,61,64,65,66,67]. Warnecke and colleagues [64] described the first SGT enzymes in the fungi *S. cerevisiae* gene *UGT51*, *C. albicans* gene *UGT51C1*, *Pichia pastoris* gene *UGT51B1*, and *Dictyostelium discoideum* gene *UGT52* using amino acid sequence similarities from a previously identified SGT enzyme (Ugt80A1, Ugt80A2) in plants, and cloned these enzymes for in vitro activity [68,69]. Nearly 15 years later the first fungal sterylglucosidase enzyme known as endoglucoceramidase-related protein 2 (EGCrP2) was identified and characterized in *C. neoformans* by Watanabe and colleagues [1]. The genetic ablation of this enzyme resulted in an accumulation of SGs as well as glucosylceramides. However, our lab also deleted this gene from *C. neoformans* as well generating the aforementioned *C. neoformans* Δ*sgl1* mutant [33]. Our mutant differed from what was observed by Watanabe et al. since this mutant was specific for only β-glucosidase activity and not physiologically relevant long chain glucosylceramides. We suggested renaming the EGCrP2 strain to *C. neoformans* Δ*sgl1,* as it has been referred to thus far. This mutant will be discussed in more detail during the immunological role of SGs with regards to fungal pathogens, as well as in the future perspectives section.

## 4. Immunomodulatory Roles of Steryl Glycosides

### 4.1. Fungi

Sphingolipids have received much attention in the immunological world, especially in terms of inflammation and stress remediation [61,70,71,72]. As was mentioned earlier, sterols, SGs, and the associated metabolic enzymes were previously studied in the immunological side of pathogenesis through exogenous addition of SGs, attenuated genetic mutant strains, upregulation of SG metabolism upon stress conditions, among other applications [3,4,15,33,61,71,73]. Before delving into the research that has been done with *C. neoformans* Δ*sgl1* mutant, a brief overview of the research leading up to this will be discussed. Since prior reviews have already covered this topic in more detail, there is no need to discuss these in major detail as only the research published in the last three years will be extensively discussed. 

From the time of the first published report on SGs from the olive plant in 1908, the first few reports implicating SGs in immunology were published in the 1960s and subtly continued up into the latter half the 1990s, when more defined studies began to appear. In 1996, Bouic and colleagues [74] used femtogram levels of the plant sterol, β-sitosterol, and the glycosylated SG, β-sitosterol glucoside, in physiologically relevant levels (100:1 mass BSS:mass BSSG) found in ~80% of higher plants. These authors reported an observed increase in T cell proliferation both in vivo, in human volunteers ingesting the found optimal concentration, and in vitro, accompanied by an upregulation of proliferative markers and Th1 cytokine secretion including interleukin-2 (IL-2) and interferon-γ (IFN-γ) with the mixed sterol:SG combinatorial treatment compared to either individual component alone. In the following year, Donald and colleagues [75] evaluated this phytosterol and SG derivative combination as adjuvant in human pulmonary tuberculosis patients. In this study, all of the patients received an anti-tuberculosis regimen, with the experimental group receiving a sitosterol adjunctive treatment and the control group receiving a placebo. The experimental group displayed a higher overall increase in body weight, increased T cell proliferation, and increased numbers of eosinophils compared to the placebo control. However, the authors can not make any definitive conclusions whether the sitosterol adjuvant was helpful since there were no differences in improved lung radiological tests or readouts in the Mantoux tests between the two groups, which implied that the sitosterol adjuvants were not necessary for a successful anti-tuberculosis regimen. However, retrospectively, these two studies along with others implicated SGs in improved T cell proliferation and activation. 

Continuing forward, two keystone studies in the mid 2000s exemplified the immunologic effect of SGs in response to disseminated Candidiasis in mice. Lee and Han [3], using ginsenoside Rg1, and Lee and colleagues [4], using daucosterol, both led to improved survival of mice pretreated with these SGs prior to infection, fewer kidney colony forming units (CFU), prolonged onset of disease, and an increase in Th1-differentiated CD4+ T cells with a subsequent decrease in Th2-based cells. The authors further confirmed that these SGs do not have any fungicidal activity, so the decrease in CFU was host-mediated control of the infection. Coinciding with the previous studies, an increase in Th1 cytokines, IL-2 and IFN-γ, was observed. However, protection was abolished when mice were administered anti-IFN-γ antibody or depleted of CD4+ T cells, which does not hold true for the necessity of protection in immunocompromised patients. Moreover, the loss of protection was restored if SG-treated CD4+ T cells were transferred into the depleted mice. Overall, these studies further confirm the use of SGs in improving the host control and outcome of the infection without any fungicidal activity. As those studies were discussed prior to this review, a more recent study utilizing SGs and *Candida albicans* reported an improvement in potential antifungal activity, as well as a membrane glucosidase hydrolyzing the compound into its active form [76]. In more detail, *C. albicans* possesses efflux pumps that continually export antifungal compounds entering the cell. Using both wild-type (WT) and efflux pump-deficient strains, the authors found that solasodine is pumped out by the WT strain but the glycosylated form, solasodine-3-*O*-β-D-glucopyranoside, exhibits antifungal activity against the WT strain and this glycosylated form is not abolished by these efflux pumps. Effectively, the glycosylated form is then further cleaved by a cytoplasmic glycosidase, revealing the active form of the compound that the authors found to have an MIC of 32 μg/ml in a previous study against several strains of *C. albicans* [62]. Studies such as these have opened up new ways of studying SGs from both a cell biology side as well as a pathogenicity approach to understanding novel antifungal therapies. Additionally, this approach reported an MIC for the SG in the study, whereas previous studies did not report or reported no MIC activity. 

In another approach not discussed in the immunological section thus far, the role of SG accumulation in a live, attenuated mutant strain of *C. neoformans* and host protection is currently under investigation in our lab. As mentioned, the Sgl1 enzyme in *C. neoformans* was determined to be a virulence factor for this yeast [33]. By limiting the concentration of SGs in the fungal cell, the host immune response is also curtailed, allowing the fungi to replicate and further dampen the immune system with other known virulence factors such as the unique polysaccharide capsule composed mainly of glucuronoxylomannan (GXM) covering the yeast [77]. The original study by Rella and colleagues [33] set the ground for the follow-up study that began to reveal the role of SGs in the complete (100%) protection, while backing up the prior findings of T cell-mediated protection, although the response was not dependent on CD4+ T cells. As an additional note, this protection was not serotype-specific as the sgl1-deficient mutant also protected against subsequent challenge with the highly virulent *Cryptococcus gattii R265* showing a robust protection. 

In the follow-up keystone study by Colombo and colleagues [2], we concluded that SGs were needed in combination with the outer GXM capsule for the observed host protection. This was accomplished using the WT H99 strain with three genetic mutants: *C. neoformans* Δ*sgl1* accumulating SGs, *C. neoformans* Δ*cap59* lacking the GXM capsule, and *C. neoformans* Δ*cap59*Δ*sgl1* an acapsular mutant accumulating SGs. The double mutant, although expressing nearly equal levels of SGs, did not provide protection in the host upon WT challenge. The double mutant did, however, get cleared from the lungs in seven days, which was a week faster than the Δ*sgl1* mutant during primary immunization [2]. This suggested two main points to us: (i) that the SGs located in the plasma membrane of the mutant may be acting as an immunoadjuvant to the GXM-based capsule, resulting in protection to future lethal challenges, and (ii) the presence of SGs mounted an immune response that may be responsible for sterilizing immunity in a mouse model. The latter point was further supported by the fact that although the Δ*cap59* mutant was avirulent, it steadily persisted for up to 60 days post-immunization without full clearance; SGs do not accumulate in this strain and were not cleared from the lungs. This lack of clearance was also observed in the Δ*gcs1* mutant strain that, although showing no virulence to the host, was never cleared from the lungs. These data both help support our hypothesis that SG accumulation yields sterilizing immunity. We do need to mention that although we checked glucosylceramide levels in these SG-accumulating strains, we cannot rule out that no other membrane lipid species may be affected, adding to this phenotype. Although one mutant strain of *C. neoformans* from another lab, *C. neoformans* Δ*apt1,* did show an accumulation of SGs to similar levels as our Δ*sgl1* strain [78], phenotypic analyses in mice would need to be confirmed before conclusions are made regarding SGs and associated virulence such as we have performed in our studies. There was reported attenuation of virulence and lack of brain colonization, but clearance would be the main goal assessed in the CBA/J mouse strain at the gender and age used in previous studies for proper comparison.

In support of the Th1-differentiated cell and cytokine responses to SGs, this study [2] also supported prior observations to the shift in the type 1 host response. Both temporally and quantitatively, *C. neoformans* Δ*sgl1* strain showed a quicker and more severe recruitment of neutrophils, dendritic cells, and CD4+ T cells, as well as pertinent type 1 and type 17 cytokines and chemokines including IL-17, IP-10/CXCL10, MIP-1⍺/CCL3, MIP-1β/CCL4, MIP2/CXCL2, and KC/CXCL1. Moreover, the Δ*sgl1* strain was observed to show a temporal resolution of infection in both the cytokine and cellular time course experiments. In other words, there was a robust cell recruitment and effector cytokine release early on upon infection, followed by a dampening of immune effectors as the timeline progressed, unlike the other strains during the vaccine efficacy challenge. Overall, this study supported the original hypothesis that GXM was necessary in combination with SGs for protection, which we now suggest was due to an immunoadjuvant effect. Although major work was uncovered using these mutant strains, there is still much work to be done as to how these SGs stimulate host responses and why they act as adjuvants in combination with the GXM-based capsule. 

Although SGs were shown in several examples to boost response to stress, not all SGs are made equally. One study investing the sterol contents of two *Kluyveromyces* strains found that *Kluyveromyces lactis* strain M-16 that produces high amounts of SGs did not have any growth advantage under high temperatures or in the presence of high concentrations of NaCl compared to *K. lactis* strain NBRC 1267 that does not naturally produce SGs [66]. The *Kluyveromyces* strains are phylogenetically similar to both *Saccharomyces cerevisiae* and *Candida albicans*, which have all been shown to produce and respond differently to SG production, even though ergosterol is the major sterol source in all of these fungi. This study as well as one studying steryl glycosyltransferase enzymes of *Pichia pastoris* and *Yarrowia lipolytica* [79] show that SGs behave differently depending on the source and effective stressors. This further promotes the use of SGs from fungi that have reported beneficial host effects, such as ones used from the *C. neoformans* Δ*sgl1* strain in our lab. 

### 4.2. Plants

As this review is meant to focus mainly on fungal pathogens and SGs, the plant and bacterial sections will be milder versions of the literature to superficially express the similarities, differences, and controversies among SG studies in these organisms compared to fungi. In plants, sterol derivatives are more ubiquitously present. Plants commonly have free sterols, steryl esters, steryl glycosides, and further derived acyl steryl glycosides, adding to the complexity and biological functions of each species, which seems fit due to the difference in immunity in plants. For a more thorough review on plant sterols and SGs, we point the reader to these more detailed reviews [15,67], as well as refer to the early studies regarding the sitosterol compounds used to combat *C. albicans* infection mentioned above. 

Sterols are a cornerstone group of compounds in plants, as the sterol base and the glyco- and acyl- conjugates are required for proper plant development, growth, and response to infection [67]. Analogous to the mammalian microbiome, plants possess an essential root-associated microbiome of epiphytic (on the root surface), endophytic (inside the root), and mycorrhizal (fungal-related) microbial symbionts that are vital to proper plant growth and metabolism. Indeed, these root-dwelling symbionts, such as the mycorrhizal fungus *Rhizophagus irregularis*, have been suggested to be necessary in proper lipid metabolism [80]. These microbes supply certain lipid species including galactolipids in plants such as *Lotus japonicus* [80]. SGs and their acyl forms have been commonly studied in other commensal mycorrhizae. Interestingly enough, although ergosterol is the major sterol associated with fungi, *R. irregularis* commonly produces 24-methylcholesterol and 24-ethylcholesterol as the major sterol sources [80], which has also been shown with other plant-associated fungal species as well. 

As was shown with fungal SGs among different genera, the same SG controversies exist in the plant realm. Two recent studies show conflicting findings regarding SGs in basal plant immunity. A 2016 study by Singh and colleagues [72] reported that silencing of sterol glycosyltransferases leads to compromised basal immunity in the plant *Withania somnifera* to the fungal pathogen *Alternaria alternata*. In other words, silencing the SGT enzyme will lower the amount of sterol being made into SG (Figure 1). As was hypothesized and shown in the fungal section, making SGs helps promote host immunity in mice during fungal infection. However, the lack of SGT enzyme (hence less SGs), decreased the growth and overall health of the plant post-infection. The authors concluded that silencing SGTs resulted in a positive feedback regulation of withanolide biosynthesis, leading to reduced biotic stress tolerance. In conflict with these results, a 2019 study by Castillo and colleagues [81] reported that inactivation of the SGT enzymes in *Arabidopsis* plants enhanced the resistance to the necrotrophic fungus *Botrytis cinerea*. The authors showed that the mutant strain of *Arabidopsis* that cannot produce SGs was more resistant to infection with *B. cinerea* due to an increase in plant stress response factors to combat the infection. The authors used mRNA transcript level readouts as well as measuring plant biomass growth and visible health. The mutant strain of *Arabidopsis* had better overall growth and outcome after infection with *B. cinerea*. There was one major critique to be made for this paper, however. The WT plant strain did not upregulate SGs as a normal response factor, so it would have been more telling to see an overexpressing strain for comparison. In conclusion however, there is the fact that SGs may act differently in different plant species and also to different plant fungal pathogens. 

### 4.3. Bacteria

The same milder version of literature regarding bacteria will also be presented here. Again, we point the reader to other reviews for more in-depth bacterial SG literature [15,82,83]. Bacteria lack the enzymes necessary for SG production [7], so the major bacteria involved in SG production literature are *Helicobacter pylori*, a human gastrointestinal pathogen, and *Borrelia burgdorferi*, the causative agent of Lyme disease, both glycosylate host cholesterol into SGs [83]. Odder, compared to the majority of the prokaryotes, *H. pylori* uses the SGs to evade the host immune system. *H. pylori* possesses a cholesterol-a-glucosyltransferase (CGT) that extracts host cholesterol and converts it to cholesteryl glucosides [45]. This is a two-pronged attack, one by depleting the host plasma membrane of cholesterol, a key constituent of lipid raft formation and phagocytosis, as well as delaying phagolysosomal maturation once phagocytosed. Since the CGT enzyme is encoded by *CapJ* in *H. pylori*, using a Δ*capj* mutant showed that faster entry into macrophages and quicker acidification and fusion of the phagolysosome was achieved compared to the WT strain. A cartoon depiction of this is clearly illustrated in Figure 6 in the paper by Du et al. [45]. 

*B. burgdorferi* also exchanges lipids with the host cell [44]. Although less in the SG realm, major sterols were shown to be actively transferred to host epithelial cells via a direct contact mechanism as well as transferred through outer membrane vesicle release. The study was conducted to investigate the (at the time) unknown roles of cholesterol glycolipids in spirochete pathogenesis. Studies such as this and earlier ones found certain lipids to have an immunogenic response. Indeed, the roles of certain glycosylated lipids decorated with various acyl groups was an important topic to the point that several reports were published on the synthesis [63] and crystal structures [84] of certain lipids and associated bacterial enzymes for increased study and characterization.

## 5. Other Known Application Uses Involving Steryl Glycosides

### 5.1. Anti-Cancer Treatments

Far fewer studies regarding sterols and SGs and anti-cancer or anti-tumorigenic therapies have been performed. Several studies in 1960s, 1970s, and 1980s investigated sitosterols in their role on reducing the carcinogen-induced cancers in rat colons (Barclay and Perdue, 1976; Hartwell and Abbott, 1969; Hartwell, 1976; Raicht et al., 1980) [74], however, the relevance of these studies at the present time does not warrant further explanation. A 2017 study [5] on the role of a novel and rare steryl ⍺-glucoside (as most SGs in the literature and discussed here are in the beta conformation) was explored in a combinatorial treatment on the MCF-7 breast cancer cell line. The authors used the *H. pylori* glycosyltransferase HP0421 enzyme construct to make this SG. Both 1 μM tamoxifen or 5 μM trans-androsteronyl-⍺-glucoside resulted in ~30% downregulation in cell viability compared to the control. When these two compounds were used in combination at the same concentrations, there was a statistically significant, additive effect on MCF-7 breast cancer cell viability by decreasing cell viability by ~62%. 

### 5.2. Biotechnology and Industrial Uses

There have been several biotechnological advancements using the SG metabolism enzymes, but not SGs themselves. These advancements span the food industry, agriculture, medicinal, and crystallization assays [6,72,73,80,85,86,87]. Three important studies showed the importance and uses of SGs in these fields. First, since SGs are made in such small amounts in cells, a means to mass produce will possibly be needed in the future, especially if our hypothesis regarding SGs as an immunoadjuvant holds true. One 2016 study reported the biochemical characterization of a steryl glycosyltransferase enzyme from the bacteria *Micromonospora rhodorangea* ATCC 31,603 [85]. This was a unique characterization since this SGT enzyme preferred phytosterols (plant sterols including sitosterol, stigmasterol, and campesterol) to cholesterol when making SGs. Since phytosterols are a common component of several other cosmetic and medicinal treatments, these enzymes have uses in the applied and biotechnological arm of science. 

On the medicinal side, SGs have also been investigated in the drug delivery system (DDS), as well. It was reported by Chang et al. [76] that the glucose moiety of the SG greatly improves uptake into cells compared to the free sterol. Originally investigated in the 1980s, SGs were used in the plasma lipoproteins as vehicles to deliver drugs or vaccines. This method of investigation continued for decades (although not too widely published), and a study by Maitani and colleagues [73] showed that SG particles facilitated drugs being delivered to the colon and increased peptide drug bioavailability after nasal and intestinal administration. Additionally, these drug delivery SGs enhanced the anti-cancer effects in liver cancer chemotherapy. 

## 6. Conclusions and Future Perspectives

With recent advancements in the knowledge of SGs and their role in immunomodulation in the host during cryptococcal infection, we strongly want to call attention to SGs, proposing these compounds warrant increased attention and further investigation as immunoadjuvants in host–fungal interactions. As SGs from different sources (plants, fungi, humans, bacteria) showed some overlapping effects, there were observable differences that also existed. The ergosterol SGs, especially from *C. neoformans*, did not show any negative effects in the outcome of the infection, so it will be advantageous to continue studies with these SGs.

It was mentioned above that the mechanisms of how and why SGs work with GXM-capsules in the host need to be uncovered to dissect this type of vaccine protection. The capsule was shown to be needed in concert with SGs for complete host protection [2], but as of now there are no studies on how SGs stimulate host immunity. There are extensive studies on how GXM modulates the host immune system looking at both upstream and downstream effects [77,88,89]. The synergistic effect of SGs with GXM is currently under investigation. This is complemented by further investigation of the combined type 1/type 17 response exhibited by hosts immunized with *C. neoformans* Δ*sgl1.* Finding mandatory effector cell populations will begin to unravel the cell-mediated protection required for this protection. This too is currently being investigated by our lab.

Aside from what has been mentioned so far, two other topics should be kept in mind when designing future experiments and hypotheses. Titan cell formation and the role of SGs found in extracellular vesicles (EVs) are important features that can be tied into how the fungal cells respond and trigger protection in the host. Titanization is the enlargement of the surrounding capsule of *C. neoformans* when grown in minimal media or the hostile host environment [90,91,92]. Since SGs are found in the plasma membrane, the multi-fold enlargement of the capsule may hinder or ablate the physical properties SGs convey to the capsule, but this is purely speculation, since this has not been investigated as well and is not currently under investigation by our group. EVs, however, were explored recently, although briefly [2], since EVs were suggested to open up a new role for lipids in fungal pathogenesis [58,93,94]. Isolating EVs from the four strains mentioned prior, these EVs were used to immunize *Galleria mellonella* worms to assess if the EVs from the protective strain also convey protection. Although sgl1-EVs prolonged death by three days, all the worms succumbed to infection and died. However, the experimental layout can possibly be optimized to assess these EVs. Multiple doses of EVs can be given to the worms, such as the vaccination strategy by Specht et al. [95,96] using glucan-based carriers. More time can also be given to assure proper host immunity. Finally, the number of EVs may have been a limiting factor or *G. mellonella* may not have been the most optimal host to test this strategy. However, with a novel protocol for the isolation of larger batches of EVs [97], testing these variations is in our sights in the future.

As suggested in the manuscript by Watanabe et al. [1], GFP-tagged SGs and/or GFP-tagged SGL1 enzymes will be paramount in studying accumulation or activity localization. Antifungal drugs designed to specifically target and block the enzymatic function of this enzyme will be a valid route from the observed data reviewed here. This is further made a possibility since the Sgl1 enzyme does not appear to have a mammalian counterpart in humans [1]. A possible problem involves the intracellular and extracellular lifestyles of *C. neoformans*, which would complicate the ability of the drug to find and enter *C. neoformans*. With these topics in mind, our lab is currently undertaking several approaches from different directions to understand this interesting and clinically relevant SG-mediated phenotype. 

## Figures and Tables

**Figure 1 jof-06-00025-f001:**
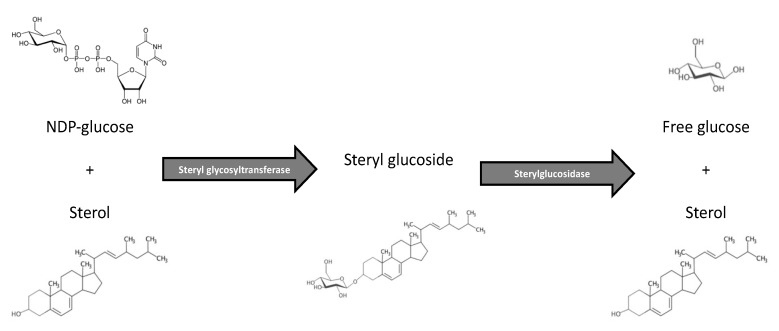
Steryl glucoside metabolism. Steryl glucosides are synthesized from a sugar moiety and free sterol via a steryl glycosyltransferase enzyme. Steryl glycosides are then hydrolyzed via a sterylglucosidase (also known as a steryl hydrolase) enzyme into its free sugar and free sterol subunits. This figure depicts the synthesis and catabolism of the steryl glucoside ergosteryl-β-3-glucoside in *Cryptococcus neoformans* using nucleotide diphosphate carrying glucose as the sugar source and free ergosterol as the sterol source. However, the number and type of sugar and the sterol species can differ by organism to make the steryl glycoside. Additionally, the anomeric bond can also be in the alpha or beta linked form. *C. neoformans* Δ*sgl1* lacks the sterylglucosidase enzyme and therefore accumulates a large concentration of steryl glucosides.

**Figure 2 jof-06-00025-f002:**
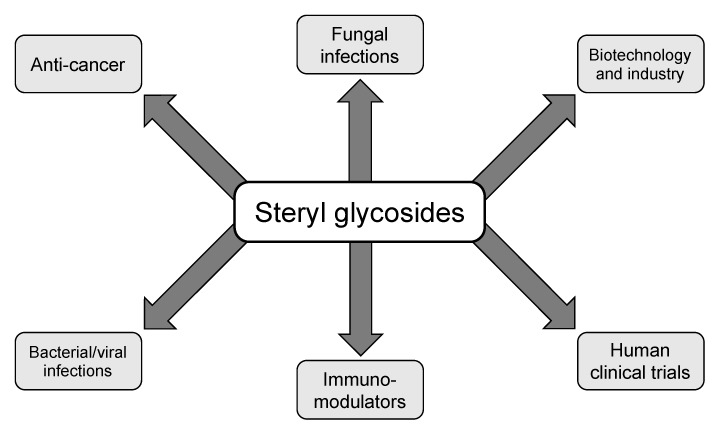
Multifaceted functions of SGs reported in the literature. Steryl glycosides have been implicated in several different aspects of biology including fungal, bacterial, and viral infections, anti-cancer treatments, immunomodulatory adjuvants, human clinical trial candidate compounds, and in the biotechnology and drug delivery system sectors. This figure is not a full representation of all uses of SGs in the literature as agricultural uses and crystallography studies have been reported, however, the graphic does pertain to the studies relevant to this review.

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
