# Peer review of "Steryl Glycosides in Fungal Pathogenesis: An Understudied Immunomodulatory Adjuvant"

_jof, 2020, doi:10.3390/jof6010025_

Round 1
Reviewer 1 Report
In this review, authors put forward the background of Steryl Glycosides in different kingdoms and their immunomodulatory role during fungal infections of animals/mammals. Authors have done a good job in giving the broad background and narrowing down to mammalian host-pathogen interactions.
The review can be accepted with following minor edits.
Authors are suggested to look into and rephrase the following sentences.
page 4 Line 23; page 6 lines 34-35; page 8 Line 28-29, lines 30-31, lines 51-52.
Author Response
In this review, authors put forward the background of Steryl Glycosides in different kingdoms and their immunomodulatory role during fungal infections of animals/mammals. Authors have done a good job in giving the broad background and narrowing down to mammalian host-pathogen interactions.
Thank you for the positive feedback regarding this work.
The review can be accepted with following minor edits.
Authors are suggested to look into and rephrase the following sentences.
All corrections to these sections are shown in bold below the original text as well as the resubmitted document.
page 4 Line 23;
Alteration of sterols is pathogenic fungi has also been explored in the past. Ghannoum and colleagues (51) studied the difference in sterol composition of C. neoformans varied in relapse compared to primary infection patients unlike C. albicans.
We edited the text to clear up the statements.
Since the sterol pathway is a common drug target in pathogenic fungi, changes in sterol composition has been explored in the past. Ghannoum and colleagues (51) investigated how the sterol composition in C. neoformans changed in clinical isolates of relapse patients compared to primary infection patients unlike what has been seen in C. albicans.
page 6 lines 34-35;
All patients were receiving an anti-tuberculosis regimen with the experimental group getting sitosterol adjunctive treatment. This group had a higher overall weight gain, increased proliferation in T cells, and higher eosinophil counts compared to the placebo control. Overall, however, no definitive conclusions were made since radiological improvements and no differences in Mantoux testing was similar between the groups.
We edited the text to clear up the statements.
In this study, all of the patients were receiving an anti-tuberculosis regimen with the experimental group receiving a sitosterol adjunctive treatment and the control group receiving a placebo. The experimental group displayed a higher overall increase in body weight, increased T cell proliferation, and increased numbers of eosinophils compared to the placebo control. However, the authors could not make any definitive conclusions if the sitosterol adjuvant was helpful since there were no differences in improved lung radiological tests or readouts in the Mantoux tests between the two groups, which implied that the sitosterol adjuvants were not necessary for a successful anti-tuberculosis regimen.
page 8 Line 28-29,
Plant sterols and all derivatives are necessary for the way plants function, develop, and response to infection (61).
We edited the text to clear up the statements.
Sterols are a cornerstone group of compounds in plants, as the sterol base and the glyco- and acyl- conjugates are required for proper plant development, growth, and response to infection (61).
lines 30-31,
Analogously to the mammalian microbiome, plants have epiphytic, endophytic, and mycorrhizal microbial symbionts that are vital to proper plant metabolism and developmental growth. Indeed, these root-dwelling symbionts, such as the mycorrhizal fungus Rhizophagus irregularis, have been suggested to be necessary in proper lipid metabolism and supply of certain species such as galactolipids in plants such as Lotus japonicus (74).
We changed the text to make the clear up our summary of the study.
Analogous to the mammalian microbiome, plants possess an essential root-associated microbiome of epiphytic (on the root surface), endophytic (inside the root), and mycorrhizal (fungal-related) microbial symbionts that are vital to proper plant growth and metabolism. Indeed, these root-dwelling symbionts, such as the mycorrhizal fungus Rhizophagus irregularis, have been suggested to be necessary in proper lipid metabolism (74). These microbes supply certain lipid species including galactolipids in plants such as Lotus japonicus (74).
lines 51-52.
The authors showed that the mutant plant not producing SG was more resistant to the fungal infection due to an increase in plant stress response factors to combat the infection. Use mRNA transcript level readouts and measuring plant growth and health, the mutant plant had a better overall growth and outcome to the infection.
We changed the text to make the clear up our summary of the study.
The authors showed that the mutant strain of Arabidopsis that could not produce SGs was more resistant to infection with B. cinerea due to an increase in plant stress response factors to combat the infection. The authors used mRNA transcript level readouts as well as measuring plant biomass growth and visible health. The mutant strain of Arabidopsis had better overall growth and outcome after infection with B. cinerea.
Reviewer 2 Report
The manuscript proposed by Normile et colleagues is an interesting paper presenting biological functions of steryl-glycosides in host-fungal pathogens. Fungal SGs have been poorly investigated. Only few studies have been described and mostly in C. neoformans.
The review is divided into four parts :
Lipids as emerging topics of study in fungal infections. In this section, authors described fungal lipids (sphingolipids, sterols) as antifungal targets and as virulence factors involved in pathogenicity. Surprisingly, authors did not mention the importance of phospholipids. Indeed, it has been shown that the Kennedy pathway is essential for the virulence in Candida albicans and putatively an antifungal target. Sterol and SGs. Here, the authors describe sterol from humans, plants and fungi. It would be appreciated to include a table summarizing the cited structures and their biological functions. Immunomodulatory roles of SGs. In this section, the authors discussed data on the beneficial effect of SGs in eradicating fungal infections, where some are modulating the host immune response and others have a fungicidal effect. Primarily, the authors discuss the role of SGs in C. neoformans as an immunoadjuvant of the capsular polysaccharide GXM that induces protection against fungal infection. This discussion is based on the recent article published by Colombo and colleagues. However, the demonstration of the adjuvant effect of SG with purified molecules remains to be demonstrated. Page 9, lines 36-37, there is no evidence that sphingolipids such as IPCs are not necessary for sterilizing immunity. The authors cannot exclude the involvement of other membrane components in host response and clearance. Please rephrase. Other known application uses involving steryl glycosides. Authors describe the potential of SGs as anti-cancer and other industrial uses.
Minor comments
Page 1 line35. Phospholipids or glycerophospholipids, which are the most abundant lipids in cellular membrane, are not indicated as lipid.
Page 3, lines 31-33. Reference is missing
Page 9, lines 37-40, In the original paper, there is no synergistic effect between tamoxifen and trans-androsteronyl-alpha-glucoside. It is only an additive effect where each compound induces around 30% decrease of cell viability. Please rephrase.
Author Response
The manuscript proposed by Normile et colleagues is an interesting paper presenting biological functions of steryl-glycosides in host-fungal pathogens. Fungal SGs have been poorly investigated. Only few studies have been described and mostly in C. neoformans.
Thank you for the interest in our work in the poorly studied topic of SGs in host-fungal pathogens.
The review is divided into four parts :
Lipids as emerging topics of study in fungal infections. In this section, authors described fungal lipids (sphingolipids, sterols) as antifungal targets and as virulence factors involved in pathogenicity. Surprisingly, authors did not mention the importance of phospholipids. Indeed, it has been shown that the Kennedy pathway is essential for the virulence in Candida albicans and putatively an antifungal target.
Although we aimed to focus this review on glycosphingolipids, namely SGs, we do find that it is important to mention some relevant background about the importance of phospholipids as this is a relevant topic in microbial pathogenesis and membrane lipids. We have included phospholipids, including the Kennedy pathway and CDP-DAG pathway, in this more generalized section of the manuscript for readers to fully understand the breadth of the field. This can be found on page 2 lines 29-41.
Sterol and SGs. Here, the authors describe sterol from humans, plants and fungi. It would be appreciated to include a table summarizing the cited structures and their biological functions.
We have contemplated a table to summarize structures and functions previously, but since none of these compounds are completely novel, we do not find this to be a necessity. There has been a recently published review (Grille et al 2010) that provides this exact summary in tabular form. Although some functions have changed or are newly discovered, we find that referring the readers to this table in that publication is sufficient. We do this on page 3 lines 22-23 ending on page 4 line 1.
Immunomodulatory roles of SGs. In this section, the authors discussed data on the beneficial effect of SGs in eradicating fungal infections, where some are modulating the host immune response and others have a fungicidal effect. Primarily, the authors discuss the role of SGs in C. neoformans as an immunoadjuvant of the capsular polysaccharide GXM that induces protection against fungal infection. This discussion is based on the recent article published by Colombo and colleagues. However, the demonstration of the adjuvant effect of SG with purified molecules remains to be demonstrated. Page 9, lines 36-37, there is no evidence that sphingolipids such as IPCs are not necessary for sterilizing immunity. The authors cannot exclude the involvement of other membrane components in host response and clearance. Please rephrase.
We revised our wording here that makes it known that this is our hypothesis, not published data. We also added text to make it clear that other lipid species could possibly be playing a role in this clearance phenomenon.
The latter point was further supported by the fact that although the Dcap59 mutant was avirulent, it persisted for up to 60 days post immunization without full clearance; no SGs accumulate in this strain and was not cleared from the lungs. This lack of clearance was also observed in the Dgcs1 mutant strain that although showed no virulence to the host, was never cleared from the lungs supporting our claim that SG-accumulation but not other sphingolipid accumulation yield sterilizing immunity.
The latter point was further supported by the fact that although the Dcap59 mutant was avirulent, it steadily persisted for up to 60 days post immunization without full clearance; SGs do not accumulate in this strain and was not cleared from the lungs. This lack of clearance was also observed in the Dgcs1 mutant strain that although showed no virulence to the host, was never cleared from the lungs. These data both help support our hypothesis that SG-accumulation yields sterilizing immunity. We do need to mention that although we checked glucosylceramide levels in these SG accumulating strains, we cannot rule out that no other membrane lipid species may be affected and adding to this phenotype.
Necessary grammatical changes were made to correct study interpretations. The changes are outlined below for the comments on pg 9 lines 37-40 section.
Other known application uses involving steryl glycosides. Authors describe the potential of SGs as anti-cancer and other industrial uses.
Minor comments
All corrections to these sections are shown in bold below the original text as well as the resubmitted document.
Page 1 line 35. Phospholipids or glycerophospholipids, which are the most abundant lipids in cellular membrane, are not indicated as lipid.
There are several classes of lipids including triacyl glycerides, sterols, sphingolipids, and glycolipids.
We have now included these two other major groups of lipid species.
There are several classes of lipids including triacyl glycerides, sterols, sphingolipids, phospholipids, glycerophospholipids, and glycolipids.
Page 3, lines 31-33. Reference is missing
Although ergosterol has been found to be the most abundant sterol species in pathogenic fungi such as C. neoformans and C. albicans as well as in non-pathogenic Saccharomyces cerevisiae, some fungi do show variations in sterol abundance such as in the subphyla Pucciniomycoina.
We have included the relevant references to the respective statements.
Although ergosterol has been found to be the most abundant sterol species in pathogenic fungi such as C. neoformans and C. albicans as well as in non-pathogenic Saccharomyces cerevisiae (10, 12), some fungi do show variations in sterol abundance such as in the subphyla Pucciniomycotina (17).
Page 9, lines 37-40, In the original paper, there is no synergistic effect between tamoxifen and trans-androsteronyl-alpha-glucoside. It is only an additive effect where each compound induces around 30% decrease of cell viability. Please rephrase.
A 2017 study (5) on the role of a novel and rare steryl ⍺-glucoside was explored in a combinatorial treatment on MCF-7 breast cancer cell line. The authors used the H. pylori glycosyltransferase HP0421 enzyme construct to achieve this. In combination with tamoxifen, there was a synergistic effect using 1 μM tamoxifen with 5 μM trans-androsteronyl-⍺-glucoside, the SG used in this study, on MCF-7 breast cancer cell viability.
Necessary grammatical changes were made to correct study interpretations.
Both 1 μM tamoxifen or 5 μM trans-androsteronyl-⍺-glucoside resulted in ~30% downregulation in cell viability compared to the control. When these two compounds were used in combination at the same concentrations, there was a statistically significant, additive effect on MCF-7 breast cancer cell viability by decreasing cell viability by ~62%.